

# Temporal inversion of the acid-base equilibrium in newborns: an observational study

Yuko Mizutani[1], Masahiro Kinoshita[2], Yung-Chieh Lin[3], Satoko Fukaya[1], Shin Kato[1], Tadashi Hisano[1], Hideki Hida[4], Sachiko Iwata[1], Shinji Saitoh[1] and Osuke Iwata[1]

[1] Department of Pediatrics and Neonatology, Nagoya City University Graduate School of Medical Sciences, Nagoya, Aichi, Japan
[2] Department of Paediatrics and Child Health, Kurume University School of Medicine, Kurume, Fukuoka, Japan
[3] Department of Pediatrics, National Cheng-Kung University, Tainan, Taiwan
[4] Department of Neurophysiology and Brain Science, Nagoya City University Graduate School of Medicine, Nagoya, Aichi, Japan

## ABSTRACT

**Background**. A considerable fraction of newborn infants experience hypoxia-ischaemia and metabolic acidosis at birth. However, little is known regarding the biological response of newborn infants to the pH drift from the physiological equilibrium. The aim of this study was to investigate the relationship between the pH drift at birth and postnatal acid-base regulation in newborn infants.

**Methods**. Clinical information of 200 spontaneously breathing newborn infants hospitalised at a neonatal intensive care centre were reviewed. Clinical variables associated with venous blood pH on days 5–7 were assessed.

**Results**. The higher blood pH on days 5–7 were explained by lower cord blood pH ($-0.131$, $-0.210$ to $-0.052$; regression coefficient, 95% confidence interval), greater gestational age (0.004, 0.002 to 0.005) and lower partial pressure of carbon dioxide on days 5–7 ($-0.005$, $-0.006$ to $-0.004$) (adjusted for sex, postnatal age and lactate on days 5–7).

**Conclusion**. In relatively stable newborn infants, blood pH drift from the physiological equilibrium at birth might trigger a system, which reverts and over-corrects blood pH within the first week of life. Given that the infants within the study cohort was spontaneously breathing, the observed phenomenon might be a common reaction of newborn infants to pH changes at birth.

Corresponding author
Osuke Iwata, o.iwata@med.nagoya-cu.ac.jp

## INTRODUCTION

The blood pH equilibrium is persistently controlled within a narrow range via innate buffers and respiratory and renal systems, and is essential in maintaining regular metabolism in human organs (*Pocock, Richards & Richards, 2017*). These systems also play important roles when the body experiences an acute derangement of the pH homoeostasis. Adults may encounter severe acute acidosis on limited occasions at critical events (*Jung et al., 2011*).
In contrast, virtually all newborn infants experience some form of hypoxia-ischaemia and acidosis due to disruption of the placental oxygen supply before the establishment of spontaneous breathing (*Gleason & Juul, 2018*). Severe birth asphyxia causes critical energy depletion and profound acidosis (*Novak, Ozen & Burd, 2018*), ultimately leading to critical events, such as hypoxic-ischaemic encephalopathy (*Douglas-Escobar & Weiss, 2015*). Relatively less severe perinatal stress may also cause serious neurological consequences. Periventricular leukomalacia is a milder form of cerebral injury in preterm infants (*Stoll et al., 2015*), the incidence of which increases with the presence of both spontaneous and induced hypocarbia (*Okumura et al., 2001*; *Stenzel et al., 2020*). A study from a large-scale cohort suggested that newborn infants who transiently required respiratory support at birth, but did not require hospitalisation, are at increased risk of developing adverse neurodevelopmental outcomes (*Odd et al., 2009*). Studies in vivo reported increased fraction of apoptotic neuronal death when cultured neurons were transiently exposed to hypoxic-ischaemic environment, and then, to alkalotic environment with sufficient oxygen and energy substrates (*Robertson et al., 2013*; *Vornov, Thomas & Jo, 1996*). Although these studies highlight the importance of understanding the biological response to the covert disruption of the pH homeostasis at birth, there currently is limited knowledge regarding foetal and neonatal reactions to the pH drift. The elucidation of such reactions and their association with the injury cascade may contribute to the development of novel strategies for the early screening, diagnosis and treatment of transition failure at birth and subsequent neurodevelopmental impairments.

We performed a retrospective observational study to test a hypothesis that the pH equilibrium at birth influences acid–base regulation within the first week of life in spontaneously breathing newborn infants.

## MATERIALS & METHODS

### Ethics approval and consent

This study was approved by the ethics committee of the Kurume University School of Medicine (H14218). The parental consent for the use of the data was not obtained since no patient identifiers were used.

### Study population

Between December 2012 and December 2015, 830 newborn infants were admitted to the neonatal intensive care unit (NICU) of Kurume University Hospital, which is a tertiary referral centre. Of these, we reviewed the clinical records of 200 newborn infants (mean $\pm$ standard deviation, 36.9 $\pm$ 1.7 weeks gestation and 2453 $\pm$ 735 g at birth), excluding 630 infants, who did not undergo blood gas analysis at birth or on days 5–7, who were still mechanically ventilated on day 7 and who were given sodium bicarbonate by day 7 ($n = 52, 472, 102$ and 4, respectively; Fig. 1).

### Sample collection

In this unit, routine blood gas analysis is performed (i) at birth (umbilical venous samples), (ii) on day 0 (venous samples typically obtained shortly after admission), on day 3

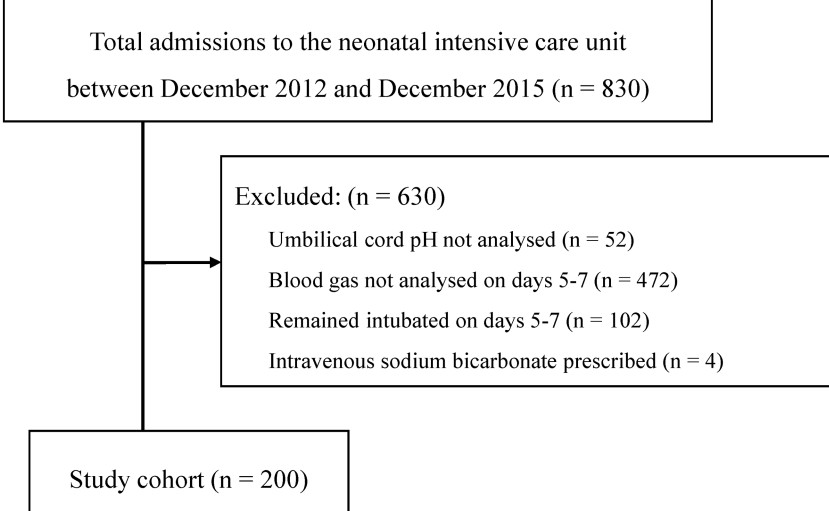

**Figure 1  Flow chart of study population.**

(venous/capillary samples) and between days 5 and 7 (venous/capillary samples obtained at the time of the newborn screening test). Blood gas analysis on days 5–7 is not performed for infants who have been discharged home or moved to the step-down unit. The blood pH; partial pressures of carbon dioxide ($pCO_2$) and oxygen; bicarbonate ($HCO_3^-$); sodium; potassium; calcium; chloride; lactate; glucose; total, carboxyl and foetal haemoglobin; and total bilirubin levels are simultaneously measured (ABL800; Radiometer, Copenhagen, Denmark). However, for cord blood samples, only the pH was available from the electronic record. Blood gas data from all routine and additional blood tests performed within 24 h of birth and between 5 and 7 days of birth were incorporated.

## Clinical variables

Clinical information was collected from the patient records, including the maternal complication, gestational age, birth weight (z-scores also calculated against the standard birth weight (*Itabashi et al., 2014*)), sex, delivery mode, Apgar scores, use of an intravenous sodium bicarbonate injection and duration of positive pressure ventilation.

## Data analysis

To highlight the selection bias, clinical backgrounds were compared between the infants within the final study cohort and those, who were excluded, using the Student's $t$-test, Mann–Whitney's U test or chi-square test. To assess temporal changes in blood pH after birth, relationships between cord blood pH and blood pH on days 5–7 were assessed adjusting for clinical backgrounds and priori covariates; blood samples obtained later than day 7 were not considered because of the selection bias derived from non-routine sampling of these samples. A generalised estimating equation was used to correct for repeated sampling and postnatal age at blood sampling (SPSS version 25; IBM, Armonk, NY, USA). Findings from the univariate analysis were not corrected for multiple comparisons;

however, *p*-values between 0.01 and 0.05 were regarded as "chance level". To evaluate the potential influence of respiratory support on the relationship between cord blood pH and blood pH on days 5–7, the univariate analysis was repeated in (i) a restrictive population of infants, who never experienced invasive respiratory support ($n = 157$) and (ii) an expanded population of the final study cohort and those, who remained intubated at the time of blood sampling on days 5–7 ($n = 302$ in total).

A multivariate model to explain the blood pH on days 5–7 was then developed for the final study cohort. First, the relationship between cord blood pH and blood pH on days 5–7 was assessed adjusting for sex and postnatal age. The influence of gestational age at birth and $pCO_2$ and lactate levels on days 5–7 was assessed using the forward stepwise selection algorithm. Values are presented as mean $\pm$ standard deviation, median [interquartile range] or number (%) unless otherwise specified.

## RESULTS

Compared with the 200 infants within the final study cohort, the 630 infants, who were excluded from the analysis, had smaller birth weight ($p = 0.008$) but statistically invariant gestational age at birth, Z-score of the birth weight, cord blood pH, 1- and 5-min Apgar scores and rates of female sex, chorioamnionitis, intubation on day 0, non-invasive positive pressure ventilation on day 0 and non-invasive positive pressure ventilation on days 5–7 (Table 1 and Supplemental Information 1). The primary indications for NICU admission for the study population were preterm birth, low-birth-weight, respiratory problems, congenital anomalies, infectious diseases, birth asphyxia, gastro-intestinal diseases, jaundice, birth trauma, other reasons ($n = 75, 35, 20, 20, 9, 4, 4, 3, 1$ and $5$, respectively) and maternal reasons (thyroid diseases, diabetes, anti-Ro/SSA autoantibodies and immune thrombocytopaenic purpura; $n = 10, 5, 5$ and $4$, respectively). A total of 157 infants never experienced invasive mechanical ventilation, whereas 43 infants were initially intubated but had been extubated by the time of blood sampling on days 5–7 (78.5% and 21.5%, respectively; Fig. 1). Non-invasive positive airway pressure ventilation was used in 27 infants on day 0 and in 17 infants at the time of blood sampling on days 5–7. On days 0 and 5–7, 453 and 249 samples were analysed, respectively (Table 1).

Univariate analysis showed that a higher blood pH on days 5–7 was associated with a greater gestational age; heavier birth weight; lower cord blood pH; higher lactate, higher glucose, higher anion gap, lower carboxyl haemoglobin and lower foetal haemoglobin levels on day 0; and lower $pCO_2$, lower calcium, lower chloride and higher $HCO_3^-$ levels on days 5–7 ($p < 0.001$, $p < 0.001$, $p < 0.001$, $p < 0.001$, $p = 0.010$, $p = 0.001$, $p = 0.001$, $p < 0.001$, $p < 0.001$, $p < 0.001$, $p = 0.001$ and $p = 0.002$, respectively; Table 2 and Online Supplemental Information 2). The relationship between cord blood pH and blood pH on days 5–7 was consistently observed in both subcohorts of infants, who never experienced invasive respiratory support (regression coefficient, $-0.154$; 95% confidence interval, $-0.258 - -0.050$; $p = 0.004$) and expanded study population including those, who remained intubated at the time of blood sampling on days 5–7 (regression coefficient, $-0.115$; 95% confidence interval, $-0.165 - -0.065$; $p < 0.001$) (Online Supplemental Information 3 and 4).

**Table 1  Background variables of the study population.**

| Variables | |
|---|---|
| Gestational age (weeks) | $36.9 \pm 1.7$ |
| Body weight at birth (g) | $2453 \pm 735$ |
| $Z$-score of the above parameter (−) | $-0.53 \pm 1.47$ |
| Female sex | 88 (44%) |
| 1-min Apgar score (−) | 8 [7–9] |
| 5-min Apgar score (−) | 9 [8–9] |
| Caesarean delivery | 96 (48%) |
| Premature rupture of the membranes | 43 (22%) |
| Hypertensive disorders of pregnancy | 14 (7%) |
| Gestational diabetes | 25 (13%) |
| Chorioamnionitis | 23 (12%) |
| Intubation on day 0 | 43 (22%) |
| Non-invasive positive pressure ventilation on day 0 | 27 (14%) |
| Non-invasive positive pressure ventilation on days 5–7[*] | 17 (9%) |
| Cord blood pH (−) | $7.29 \pm 0.09$ |
| Requirement for phototherapy | 89 (45%) |
| Postnatal age at full enteral nutrition >100 ml/kg (days) | $4.2 \pm 3.6$ |
| Blood tests on day 0 | |
| pH (−) | $7.30 \pm 0.11$ |
| $pCO_2$ (mmHg) | $48.0 \pm 13.6$ |
| $HCO_3^-$ (mmol/L) | $22.3 \pm 3.3$ |
| Lactate (mmol/L) | $0.42 \pm 0.36$ |
| Glucose (mg/dL) | $86 \pm 47$ |
| $Na^+$ (mmol/L) | $136 \pm 3$ |
| $K^+$ (mmol/L) | $4.8 \pm 1.0$ |
| $Ca^{2+}$ (mmol/L) | $1.26 \pm 0.12$ |
| $Cl^-$ (mmol/L) | $106 \pm 4$ |
| Anion gap (mmol/L) | $7.4 \pm 5.2$ |
| Total haemoglobin (g/dL) | $17.4 \pm 3.2$ |
| Carboxyl haemoglobin (%) | $1.4 \pm 0.7$ |
| Foetal haemoglobin (%) | $77.3 \pm 11.4$ |
| Total bilirubin (mg/dL) | $2.3 \pm 1.6$ |
| Blood tests on days 5–7 | |
| pH (−) | $7.37 \pm 0.05$ |
| $pCO_2$ (mmHg) | $41.3 \pm 6.4$ |
| $HCO_3^-$ (mmol/L) | $23.0 \pm 2.9$ |
| Lactate (mmol/L) | $0.22 \pm 0.09$ |
| Glucose (mg/dL) | $90 \pm 21$ |
| $Na^+$ (mmol/L) | $138 \pm 4$ |
| $K^+$ (mmol/L) | $4.7 \pm 0.8$ |
| $Ca^{2+}$ (mmol/L) | $1.24 \pm 0.14$ |
| $Cl^-$ (mmol/L) | $108 \pm 5$ |

**Table 1** (*continued*)

| Variables | |
|---|---|
| Anion gap (mmol/L) | $7.6 \pm 2.5$ |
| Total haemoglobin (g/dL) | $16.7 \pm 2.3$ |
| Carboxyl haemoglobin (%) | $1.2 \pm 0.5$ |
| Foetal haemoglobin (%) | $73.1 \pm 13.4$ |
| Total bilirubin (mg/dL) | $8.9 \pm 3.5$ |
| Heart rates on day 0 (beats per min) | $135 \pm 13.2$ |
| Respiratory rates on day 0 (breaths per min) | $47.5 \pm 10.2$ |
| Heart rates on days 5–7 (beats per min) | $142.0 \pm 11.1$ |
| Respiratory rates on days 5–7 (breaths per min) | $44.8 \pm 7.5$ |

**Notes.**

Values are shown as mean $\pm$ standard deviation, median [interquartile range] or number (%).

*At the time of blood sampling.

The first multivariate model (Model 1) was adjusted for the postnatal age and sex, which showed a negative relationship between the cord blood pH and blood pH on days 5–7 ($p < 0.001$, Table 3). Model 2 included the variables in Model 1 and gestational age, whereas Model 3 included the variables in Model 2 plus $pCO_2$ on days 5–7, both of which supported the relationship between the cord blood pH and blood pH on days 5–7 ($p = 0.005$ and $p < 0.001$, respectively; Table 3). The final model (Model 4) was adjusted for the same variables as Model 3 plus the lactate level on days 5–7, where lower cord blood pH, greater gestational age and lower $pCO_2$ levels on days 5–7 (regression coefficient: $-0.131$, $0.004$ and $-0.005$; 95% confidence interval: $-0.210$ to $-0.052$, $0.002$ to $0.005$ and $-0.006$ to $-0.004$; and $p = 0.001$, $p < 0.001$ and $p < 0.001$, respectively) explained the higher venous blood pH on days 5–7 (Table 3). To identify the intermediate variable explaining the relationship between the cord blood pH and blood pH on days 5–7, Model 5 was additionally adjusted for the covariates in Model 4 plus $HCO_3^-$ on days 5–7, where the role of cord blood pH as an independent variable for blood pH on days 5–7 was replaced by $HCO_3^-$ on days 5–7 ($p < 0.001$, Table 3)

## DISCUSSION

In spontaneously breathing newborn infants, a higher blood pH on days 5–7 was paradoxically related to a lower cord blood pH. Considering that the pH equilibrium of these infants was likely to be determined by their own spontaneous regulation, this temporal inversion of the blood pH homeostasis within the first week of life might represent a physiological response in newborn infants to the drift in the pH equilibrium at birth.

In our current study, higher blood pH levels on days 5–7 were best explained by the cord blood pH, as well as gestational age, $pCO_2$ and lactate levels of the same blood sample obtained on days 5–7. Considering that the role of the cord blood pH as an independent variable for the blood pH on days 5–7 was superseded by $HCO_3^-$ levels on days 5–7, it would be relevant to speculate that the pH drift in the cord blood towards an acidic (alkalotic) equilibrium triggered the active accumulation (elimination) of blood $HCO_3^-$ to temporally invert the acid–base homeostasis within the first week of life. Although our study

**Table 2  Dependence of the blood pH on days 5–7 on the clinical variables: univariate analysis.**

| Variables | | Regression coefficient | | | *p* |
|---|---|---|---|---|---|
| | | Mean | 95% confidence interval | | |
| | | | Lower | Upper | |
| Gestational age (weeks) | | 0.005 | 0.004 | 0.007 | **<0.001** |
| Body weight at birth (per 100 g) | | 0.002 | 0.001 | 0.003 | **<0.001** |
| *Z*-score of the above parameter | | 0.001 | −0.004 | 0.006 | 0.643 |
| Female sex | | 0.001 | −0.014 | 0.017 | 0.848 |
| Cord blood pH | | −0.126 | −0.180 | −0.073 | **<0.001** |
| 1-min Apgar score | | 0.002 | −0.005 | 0.005 | 0.879 |
| 5-min Apgar score | | −0.002 | −0.007 | 0.003 | 0.471 |
| Blood tests on day 0 | Age in hour | −0.001 | −0.001 | 0.001 | 0.850 |
| | pH | −0.070 | −0.153 | 0.013 | 0.100 |
| | $pCO_2$ (mmHg) | 0.000 | 0.000 | 0.001 | 0.147 |
| | $HCO_3^-$ (mmol/L) | −0.002 | −0.005 | 0.001 | 0.125 |
| | Lactate (mmol/L) | 0.034 | 0.017 | 0.050 | **<0.001** |
| | Glucose (mg/dL) | 0.000 | 0.000 | 0.000 | 0.010 |
| | $Na^+$ (mmol/L) | 0.001 | −0.001 | 0.003 | 0.347 |
| | $K^+$ (mmol/L) | −0.005 | −0.013 | 0.003 | 0.238 |
| | $Ca^{2+}$ (mmol/L) | 0.043 | −0.013 | 0.099 | 0.131 |
| | $Cl^-$ (mmol/L) | −0.002 | −0.004 | 0.000 | 0.074 |
| | Anion gap (mmol/L) | 0.002 | 0.007 | 0.004 | **0.001** |
| | Total haemoglobin (g/dL) | 0.000 | −0.003 | 0.008 | 0.929 |
| | Carboxyl haemoglobin (%) | −0.018 | −0.029 | −0.008 | **0.001** |
| | Foetal haemoglobin (%) | −0.001 | −0.002 | −0.001 | **<0.001** |
| | Total bilirubin (mg/dL) | −0.002 | −0.007 | 0.002 | 0.330 |
| Blood tests on days 5–7 | Postnatal age (days) | 0.013 | −0.002 | 0.027 | 0.085 |
| | $pCO_2$ (mmHg) | −0.005 | −0.006 | −0.004 | **<0.001** |
| | $HCO_3^-$ (mmol/L) | 0.005 | 0.002 | 0.008 | **0.002** |
| | Lactate (mmol/L) | 0.032 | −0.035 | 0.099 | 0.346 |
| | Glucose (mg/dL) | 0.000 | −0.001 | 0.000 | 0.173 |
| | $Na^+$ (mmol/L) | 0.000 | −0.003 | 0.002 | 0.754 |
| | $K^+$ (mmol/L) | −0.001 | −0.011 | 0.009 | 0.847 |
| | $Ca^{2+}$ (mmol/L) | −0.098 | −0.152 | −0.044 | **<0.001** |
| | $Cl^-$ (mmol/L) | −0.003 | −0.004 | −0.001 | **0.001** |
| | Anion gap (mmol/L) | 0.002 | −0.001 | 0.005 | 0.120 |
| | Total haemoglobin (g/dL) | 0.001 | −0.002 | 0.005 | 0.448 |
| | Carboxyl haemoglobin (%) | 0.015 | −0.030 | 0.033 | 0.110 |
| | Foetal haemoglobin (%) | 0.000 | −0.001 | 0.000 | 0.163 |
| | Total bilirubin (mg/dL) | 0.003 | 0.001 | 0.005 | 0.012 |
| Heart rates on day 0 (beats per min) | | −0.001 | −0.001 | 0.000 | 0.091 |
| Respiratory rates on day 0 (breaths per min) | | 0.000 | −0.001 | 0.000 | 0.420 |
| Heart rates on days 5–7 (beats per min) | | −0.001 | −0.002 | 0.000 | 0.071 |
| Respiratory rates on days 5–7 (breaths per min) | | 0.000 | −0.001 | 0.001 | 0.985 |

**Notes.**
Statistical significance was assumed for $p < 0.01$ (indicated in bold).

**Table 3  Dependence of the blood pH on days 5–7 on the cord blood pH: multivariate models.**

| Variables | Regression coefficient | | | p |
| --- | --- | --- | --- | --- |
| | Mean | 95% confidence interval | | |
| | | Lower | Upper | |
| **Model 1** | | | | |
| Cord blood pH | −0.127 | −0.181 | −0.073 | **<0.001** |
| Postnatal age (days) | 0.014 | 0.000 | 0.027 | **0.047** |
| Female sex | −0.001 | −0.015 | 0.013 | 0.845 |
| **Model 2** | | | | |
| Cord blood pH | −0.088 | −0.149 | −0.027 | **0.005** |
| Postnatal age (days) | 0.007 | −0.006 | 0.020 | 0.283 |
| Female sex | −0.001 | −0.014 | 0.012 | 0.866 |
| Gestational age (weeks) | 0.005 | 0.003 | 0.006 | **<0.001** |
| **Model 3** | | | | |
| Cord blood pH | −0.120 | −0.184 | −0.055 | **<0.001** |
| Postnatal age (days) | −0.001 | −0.012 | 0.010 | 0.830 |
| Female sex | −0.002 | −0.011 | 0.008 | 0.757 |
| Gestational age (weeks) | 0.004 | 0.002 | 0.005 | **<0.001** |
| $pCO_2$ on days 5–7 (mmHg) | −0.005 | −0.005 | −0.004 | **<0.001** |
| **Model 4 (Final model)** | | | | |
| Cord blood pH | −0.131 | −0.210 | −0.052 | **0.001** |
| Postnatal age (days) | −0.001 | −0.011 | 0.009 | 0.844 |
| Female sex | −0.001 | −0.011 | 0.009 | 0.888 |
| Gestational age (weeks) | 0.004 | 0.002 | 0.005 | **<0.001** |
| $pCO_2$ on days 5–7 (mmHg) | −0.005 | −0.006 | −0.004 | **<0.001** |
| Lactate on days 5–7 (mmol/L) | −0.006 | −0.013 | 0.000 | 0.058 |
| **Model 5 (Additional model)** | | | | |
| Cord blood pH | 0.016 | −0.003 | 0.035 | 0.092 |
| Postnatal age (days) | 0.001 | −0.001 | 0.005 | 0.296 |
| Female sex | 0.002 | −0.001 | 0.005 | 0.225 |
| Gestational age (weeks) | 0.001 | 0.000 | 0.001 | 0.118 |
| $pCO_2$ on days 5–7 (mmHg) | −0.009 | −0.010 | −0.008 | **<0.001** |
| Lactate on days 5–7 (mmol/L) | 0.001 | 0.000 | 0.003 | 0.149 |
| $HCO_3^-$ on days 5–7 (mmol/L) | 0.016 | 0.014 | 0.018 | **<0.001** |

**Notes.**
Statistical significance was assumed for $p < 0.05$ (indicated in bold).

cohort did not include those with severe birth asphyxia or excessive immaturity and all of the infants who were included had been weaned from mechanical ventilation by the time of blood gas analysis on days 5–7, even transient resuscitation and mechanical ventilation may well influence the acid–base regulation thereafter. However, the relationship between cord blood pH and blood pH on days 5–7 was consistently observed in both restrictive (those who never experienced invasive respiratory support) and expanded (the original cohort plus those who remained intubated at the time of blood sampling on days 5–7) study populations, suggesting that the influence of the respiratory support on the temporal

change of the blood pH may be limited. Provided that the blood pH equilibrium of these infants was determined as a consequence of spontaneous acid–base control, the negative relationship between the cord blood pH and the blood pH on days 5–7 might be a physiological response in newborn infants.

In foetuses, acidosis is compensated for by innate buffers, such as bicarbonate and haemoglobin, which help eliminate carbon dioxide via the placenta (*Blechner, 1993*). After birth, the role of the placenta in maintaining the acid–base homeostasis is replaced by the lung and proximal tubule of the kidney, which, together with the innate buffering system, contribute to ameliorate acidosis. Proton transporters, such as the $Na^+/H^+$ exchanger and proton ATPase, play a central role in the pH homeostasis by the proximal tubule (*Hamm, Nakhoul & Hering-Smith, 2015*; *Pirojsakul et al., 2015*). The $Na^+/H^+$ exchanger is also involved in the maintenance of the intracellular pH and cell volume in a range of cell types (*Uria-Avellanal & Robertson, 2014*). Severe birth asphyxia triggers anaerobic glycolysis, leading to the accumulation of lactate within the intracellular fluid (*Rainaldi & Perlman, 2016*). The acidic shift of the intracellular fluid activates the function of the $Na^+/H^+$ exchanger. Interestingly, once this transporter is activated by severe hypoxia-ischaemia, elimination of intracellular protons persists even after the intracellular pH is normalised, leading to an intracellular alkaline overshoot, or a pH paradox, in the neuronal tissue of asphyxiated newborn species (*Kendall et al., 2006*; *Robertson et al., 2005*; *Robertson et al., 2002*; *Uria-Avellanal & Robertson, 2014*). The pH paradox appears to be relevant as a homeostatic reaction to acidosis. However, this phenomenon is simultaneously linked to cell death and severe tissue injury, rather than survival (*Robertson et al., 2002*; *Uria-Avellanal & Robertson, 2014*). A similar phenomenon to the pH paradox has been observed in the pH control of the extracellular fluid. Helmly et al. showed that during the recovery phase from hypoxia-ischaemia, the seizure burden is tightly linked with an alkaline overshoot recovery of the extracellular pH (*Helmy et al., 2012*). These findings suggest that an alkaline overshoot of the intracellular and extracellular fluids within the cerebral tissue might be a common consequence of severe acidosis and may play an important role in the progression of cerebral injury. Although we did not assess pH levels of the cerebral tissue and cerebrospinal fluid, a similar reaction to the pH paradox might exist for the tubular $Na^+/H^+$ exchanger (*Ibrahim, Lee & Curthoys, 2008*; *Twombley et al., 2010*), which might be activated to correct and reverse the blood pH equilibrium in response to even a mild acidic drift in the blood at birth.

An alternative explanation is possible from the spontaneous down-regulation of energy metabolism and thermogenesis following severe hypoxia–ischaemia, observed in a range of vertebrates (*Wood & Gonzales, 1996*; *Wood et al., 2018*). Downregulation of energy metabolism may cause an alkaline shift in the pH equilibrium via reduced production of carbon dioxide and lactate. Indeed, an alkaline shift in the blood pH has been observed in newborn infants who were managed with therapeutic hypothermia, although most of these findings are accompanied by reduced blood carbon dioxide levels (*Pappas et al., 2011*; *Szakmar et al., 2018*; *Thoresen, 2008*).

Several limitations in the current study need to be addressed. First, the $pCO_2$, $HCO_3^-$ and lactate levels were not available from the cord blood gas analysis, leading to uncertainty

in which type of pH drift at birth triggered the inversion of pH homeostasis thereafter. Second, as previously described, our study cohort was of NICU infants. Although it is ethically unacceptable to perform serial blood sampling in healthy newborn infants, our findings need to be reassessed in a term-born cohort with relatively more mature tubular and respiratory functions, acid–base regulation system and homogeneous clinical backgrounds. Finally, because our current findings are not similar to previously reported phenomena in the clinical setting, the proposed explanations of the temporal inversion of pH equilibrium after birth are only speculative, and the precise mechanism needs to be elucidated by future studies.

## CONCLUSIONS

An acidic shift in the blood pH at birth led to an increase in the bicarbonate and blood pH on days 5–7 in spontaneously breathing NICU infants. To our knowledge, this is the first study to report a temporal inversion of the blood pH within the first week of life. Further investigation of this phenomenon may help reveal a novel injury cascade and physiological response to the pH drift at birth in newborn infants.

### Funding

This work was supported by the Japan Society for the Promotion of Science (Grants-in-Aid for Scientific Research 20H00102, 16K09005 and 18K07795). The funders had no role in study design, data collection and analysis, decision to publish, or preparation of the manuscript.

### Grant Disclosures

The following grant information was disclosed by the authors:
Japan Society for the Promotion of Science (Grants-in-Aid for Scientific Research): 20H00102, 16K09005, 18K07795.

### Competing Interests

The authors declare there are no competing interests.

### Author Contributions

- Yuko Mizutani and Osuke Iwata conceived and designed the experiments, performed the experiments, analyzed the data, prepared figures and/or tables, authored or reviewed drafts of the paper, and approved the final draft.
- Masahiro Kinoshita, Satoko Fukaya, Shin Kato, Tadashi Hisano, Hideki Hida and Shinji Saitoh performed the experiments, prepared figures and/or tables, authored or reviewed drafts of the paper, and approved the final draft.
- Yung-Chieh Lin and Sachiko Iwata performed the experiments, analyzed the data, prepared figures and/or tables, authored or reviewed drafts of the paper, and approved the final draft.

## Human Ethics

The following information was supplied relating to ethical approvals (i.e., approving body and any reference numbers):

This study was approved by the ethics committee of the Kurume University School of Medicine (H14218).

## Data Availability

The raw data are available in the Supplemental File.

## Supplemental Information

Supplemental information for this article can be found online at http://dx.doi.org/10.7717/peerj.11240#supplemental-information.

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
