# Peer review of "Temporal inversion of the acid-base equilibrium in newborns: an observational study"

_PeerJ, doi:10.7717/peerj.11240_

## Round 0.1 · original submission · Major Revisions

The reviewers have indicated several issues which you should sort out in a revised version of the text.

Reviewer 1 ·

Basic reporting

no comment

Experimental design

no comment

Validity of the findings

no comment

Additional comments

To the authors

In this manuscript, Dr. Mizutani Y and her colleagues conducted an observational study to investigate the relationship between the pH drift at birth and postnatal acid-base regulation within the first week of life in spontaneously breathing newborn infants. They founded that the higher blood pH on days 5-7 were significantly correlated with lower cord blood pH, greater gestational age, lower partial pressure of carbon dioxide and lower lactate level, with adjustment for sex and postnatal age.
This manuscript might bring a novel perspective for the treatment strategy of transition failure at birth and subsequent neurodevelopment impairments. However, to accept the manuscript, there are still concerns about description that the authors should correct or give additional comments on.

Major problems
1. Study population
 Although you investigated it by using spontaneously breathing infants in the study, you study population included 43 infants with initially ventilated infants at birth. If you like to see the biological response to the pH drift from the physiological equilibrium, why didn't you exclude them and just focus on 153 non-ventilated infants? Especially if you want to move on your next study based on the current study results, then you had better do so.
 According to the cord blood pH, the severity of general status of your study population did not seem to be less, because they were almost within normal ranges. Therefore, I am still thinking that it would be better to look at the same parameter in severe cases, as well, to clearly show the postnatal acid-base regulation in both normal and abnormal conditions.
2. Rationale of investigating blood pH on d5-7
Could you please give some comments on the rationale of investigating blood pH especially on d5-7.
3. Trend in pH from birth to d5-7
I am curious about the trends in pH from birth to d5-7, not just two point of pH, at birth and d5-7. To show the detail of biological response, those information might be more informative for understanding of biological response to the covert disruption of pH homeostasis at birth.

Minor problems
1. How about Anion Gap in the blood gas analysis?
2. As you’ve already noted in Discussion, pleases clarify the timing of taking blood gas sample of PCO2 and lactate, here in Abstract or Table3, too.
3. In discussion, your bibliographic consideration looks great, but there seems to be too much speculation. You mention about an intracellular alkaline overshoot for severe acidosis, but in your study population, their pH did not seem to show any excess acidemia at birth or excess alkaline shift at d5-7.

Reviewer 2 ·

Basic reporting

- I recommend authors provide more details regarding the effect of pH drift on neurodevelopmental impairments in Line 64
- Issue in line 72: dot and parenthesis are separated by a space.
- Additional space in line 103 between “pH” and “on”
- More information is required for how the multivariate model was developed in Line 107
- Line 108: what is the correlation? Is it statistically significant?
- Line 112: Again, information regarding the developed models is missing. It is necessary to clarify the details.
- Table 1 is somewhat confusing specially for those working in a different field. For example, the values in the table have brackets/parenthesis in front of them without any description about what they mean. In addition, there are some variables reported with no unit, it is highly recommended to use “(-)” if the variable is dimensionless.
- Line 127-148: All major findings are reported without in-depth explanation and the results are summarized in tables (Table 2 and 3). I highly recommend considering adding a better visualization as simple as a 2D graph instead of reporting numbers that are not easy to compare across different variables and conditions.

Experimental design

- This study lacks a solid hypothesis about the processes and factors involved in blood pH regulation of newborns.

Validity of the findings

no comment

Additional comments

- This is an interesting observation about blood pH regulation found to happen within the first week of life. However, I believe including graphical representation of the findings will enable readers to understand and digest this article easier. Moreover, a comprehensive discussion is necessary on the possible underlying processes that control the pH regulation during the first week of life. Finally, a detailed description on the models developed is missing that needs to be addresses.

Reviewer 3 ·

Basic reporting

Cluttered, incoherent with no emphasis on the salient points. Too difficult to follow with no correlation to the clinical conditions of the babies categorised to the reasons they were admitted in the first place. These infants are not representative of the normal population unless their underlying conditions are clearly spelt out.

Experimental design

Flawed. These are not normal spontaneously breathing infants but were sick and on the road to recovery. Spontaneously breathing infants not mechanically ventilated do not equate healthy infants. I find it difficult to accept the conclusion when there was no mention at all about the underlying disease particularly any residual lung disease, the respiratory rate, temperature - these could impact on hyperventilation, low CO2 and therefore higher pH in the sicker infants initially who are recovering. These cannot be considered to represent the normal "over-correction" phenomenon claimed by the authors.
The authors need to specify what were the clinical conditions of these infants, the average respiratory rates and temperatures at the D5-7 sampling day.
As this was an observational study, all the other infants excluded should actually be included for their data to be analysed according their individual conditions.
Why were the 4 who received bicarbonate be excluded. We should look at these infants and examine why was bicarbonate required, which is seldom given nowadays anyway in neonatal care.

Validity of the findings

Without correlation with the underlying clinical conditions, the blood investigation results could not be interpreted sufficiently accurately. The conclusion also could not be validated and confirmed as such.

Additional comments

It is crucial that the clinical diagnosis of the 200 infants be spelt out clearly from the beginning, i.e.
what were they admitted for - severe lung disease, or no lung disease but suspected bacterial infection, feeding difficulties, asphyxia? etc.
Some of these infants were actually on respiratory support at D5-7, which will not represent the normal babies acid-base status as such.
The heterogeneity of this population makes it difficult for the conclusions to be drawn accurately.
Other information need to be spelt out includes : were the babies on intravenous fluid plus enteral feeds at D5-7. Were there babies who were receiving phototherapy, and what were their hydration status as such.
The voluminous metabolic data need to be correlated with the clinical status of the babies for accurate conclusions to be drawn.

---

## Round 0.2 · accepted · Accept

All the reviewers' concerns have been correctly addressed for the authors in this new version of the text.

Reviewer 1 ·

Basic reporting

no comment

Experimental design

no comment

Validity of the findings

no comment

Additional comments

Thank you for your revised manuscript entitled “Temporal inversion of the acid-base equilibrium in newborns: an observational study”
The revised manuscript shows much improvement. I appreciate your complete response to my comments and accept it.